# Physicochemical Properties of Two Generations of MTA-Based Root Canal Sealers

**DOI:** 10.3390/ma14205911

**Published:** 2021-10-09

**Authors:** Sawsan Abu Zeid, Hadeel Yaseen Edrees, Abeer Abdulaziz Mokeem Saleh, Osama S. Alothmani

**Affiliations:** 1Department of Endodontics, Faculty of Dentistry, King Abdulaziz University, Jeddah 21589, Saudi Arabia; hedrees@kau.edu.sa (H.Y.E.); aasaleh@kau.edu.sa (A.A.M.S.); osalothmani@kau.edu.sa (O.S.A.); 2Department of Endodontics, Faculty of Dentistry, Cairo University, Cairo 12613, Egypt

**Keywords:** MTA-based root canal sealer, physical and chemical properties, pH, solubility, releasing element

## Abstract

This study evaluated the physicochemical properties and the effect of solubility on the surface morphology and composition of the root canal sealers MTA-Bioseal, MTA-Fillapex, and Adseal. Discs (*n* = 10) of freshly mixed sealer were prepared and then analyzed by Fourier transform infrared (FTIR) spectroscopy and scanning electron microscopy/energy-dispersive X-ray spectroscopy (SEM/EDX). The discs were immersed for 1, 7, 14, and 28 days in deionized water. The solubility %; pH change of the solution; and released calcium, phosphate, and silicon were measured for each period. The flowability and film thickness were also evaluated. Changes in the surface morphology and composition after 28 days of immersion were evaluated by SEM/EDX. The data were statistically analyzed by one-way ANOVA at *p* < 0.05. The FTIR and EDX results revealed similar compositions of MTA-Bioseal and MTA-Fillapex, but with different concentrations. The two MTA-based sealers had higher solution alkalinity (pH > 10) than Adseal (pH ≈ 8.5). MTA-Fillapex exhibited the highest solubility % and the largest calcium and silicon ion release. MTA-Bioseal had the highest phosphate ion release. After 28 days, the sealer surfaces showed large micropores, with larger pores in MTA-Fillapex. Adseal had an intermediate flowability but exhibited the greatest film thickness. Finally, the highest solubility and largest amount of silicon release was exhibited by MTA-Fillapex, which might predispose it to the development of large micropores, compromising the apical seal of obturation.

## 1. Introduction

Fostering a fluid-tight apical seal throughout the root filling after instrumentation is crucial for a favorable long-term outcome of root canal treatment. Because gutta-percha lacks adhesiveness, a root canal sealer must be used to fill the minute spaces between the gutta-percha and the canal wall to provide a three-dimensional seal of the root canal system [1].

A wide variety of root canal sealers is commercially available, and these possess different compositions and physicochemical properties. A sealer’s performance depends on its composition and physical and chemical properties. Owing to the favorable cohesive strength, biological behavior, and osteogenic potential of mineral trioxide aggregate (MTA) [2], several MTA-based root canal sealers have been formulated to utilize these advantages. MTA-Fillapex was the first generation of MTA-based root canal sealer and was launched by Angelus (Angelus, Londrina, Brazil) in 2010 as a paste–paste formulation [3]. According to the suggested ideal requirements for an endodontic sealer proposed by Grossman [4], to achieve three-dimensional obturation, the sealer should be insoluble in tissue fluid, inhibit bacterial growth, and properly flow along the dentinal tubules when it is first applied. In previous studies, MTA-Fillapex showed a high solubility that exceeded the acceptable limit [5,6]. Another study showed that MTA-Fillapex was cytotoxic for 4 weeks after its application, which was attributed to its high dissolution rate [7].

A new MTA-based sealer, MTA-Bioseal, has been recently introduced by ITENA Clinical (Paris, France). The manufacturer claims it exhibits limited expansion during setting, low solubility when it contacts tissue fluid, and optimal flowability [8]. However, no available studies have reported its physical and chemical properties.

The current study evaluated the physicochemical properties (solubility %; pH changes; released calcium, phosphate, and silicon ions; flowability; and film thickness) and the effect of solubility on surface morphology and composition of two generations of MTA-based root canal sealers (MTA-Fillapex and MTA-Bioseal) and compared them with an epoxy resin-based sealer (Adseal, META Biomed Co., Chungbuk, Korea), which was considered as a control. The null hypothesis was that there would be no significant difference among the tested sealers for any of the parameters assessed.

## 2. Materials and Methods

The study design and protocol were approved by the Ethics Committee of the Faculty of Dentistry, King Abdulaziz University (#216-01-21).

### 2.1. Sample Preparation

According to ISO 6876 and ANSI/ADA Specification No. 57 for root canal filling [9,10], a fresh mix of each sealer was prepared in accordance with the manufacturer’s instructions and inserted into a polyethylene mold (10 mm diameter, 3 mm high). Discs (*n* = 10/sealer) were wrapped with moistened gauze and incubated at 37 °C and 100% humidity until material hardening.

### 2.2. Fourier Transform Infrared (FTIR) Analysis

One disc of each sealer was analyzed by FTIR spectroscopy (Vertex 80v, Bruker, Karlsruhe, Germany) to determine the composition. The spectra were obtained at 4000–400 cm^−1^ and 4 nm resolution.

### 2.3. Solubility

After complete hardening, each disc was weighed (W0) using an electric balance (#ZSA210, Scientech, Boulder, CO, USA), placed in a vial containing 10 mL of deionized water, and incubated at 37 °C and 100% humidity. After each immersion period (1, 7, 14, and 28 days), the discs were removed and dried on blotting paper overnight, then reweighed (Wt1, Wt7, Wt14, and Wt28). The solubility percentage (%) was calculated by the following Equation (1) [11]:(1)$$\mathrm{Solubility}\;\%=\frac{W0-Wt1}{W0}\times 100_{\;}$$

### 2.4. pH Changes

After each immersion period, the solution was evaluated for pH changes at 25 °C using a pH meter (Jenway 3510 pH meter, Bibby Scientific Ltd., Stone, UK) initially calibrated with standard pH 4.0 and 7.0 solutions [5].

### 2.5. Released Elements

After each immersion period, the deionized water was analyzed for the amount of released calcium (Ca^2+^), phosphorus (P^3−^), and silicon (Si^4+^) ions, which were respectively analyzed using an EDTA titration method [12], a colorimetric method with a spectrophotometer (Jenway 6705 UV/Vis spectrophotometer, Stone, UK) [13,14], and inductively coupled plasma optical emission spectroscopy (Agilent 5100, Santa Clara, CA, USA).

### 2.6. Scanning Electron Microscopy (SEM) and Energy-Dispersive X-Ray (EDX) Analysis

The set discs were analyzed by SEM/EDX (Octane pro, 7.2/15252, EDAX, Ametek Material Analysis Division, Mahwah, NJ, USA) to determine the surface morphology and composition of each sealer before immersion in deionized water. At the end of the final immersion period (i.e., after 28 days), the discs were reexamined to determine the surface and composition changes consequent to solubility. The microporosities in each image were measured using ImageJ software, a Java-based image processing program, (version 1.44, 64-bit Java 1.8.0_112, National Institutes of Health, Bethesda, MD, USA).

### 2.7. Flowability and Film Thickness

The flowability test was conducted based on ISO 6876/2001 for dental root canal sealing material [10]. One drop of 0.05 ± 0.005 volumes of each mixed sealer (*n* = 5) was applied onto a glass slab (35 × 35 × 6 mm^3^) [15]. After 3 min, it was covered by another glass slab weighing 20 mg, and an additional weight of 100 g was placed on the top of the spreading sealer. The two glass slabs containing the sealer and the 100 g weight were incubated for 10 min at 37 °C and 100% humidity. After removing the weight and the upper glass slab, the dimensions of the circular sample were measured using a digital caliper (Cole-Parmer Canada Inc., Montreal, QC, Canada). In cases where the obtained circle was not uniform or if the dimension exceeded 1 mm, the test was repeated.

After finishing the flowability test, the thickness of the double slab containing the set sealer (Ts) was measured by a digital caliper. The thickness of an empty double slab (T0) was also measured. The sealer film thickness was calculated as Ts–T0.

### 2.8. Statistical Analysis

The recorded data (solubility %; pH; released Ca^2+^, PO_4_^3−^, and Si^4+^; and EDX) were statistically analyzed by one-way ANOVA and the post hoc Tukey HSD test using SPSS software (version 20.0; SPSS, Inc., Chicago, IL, USA). Comparisons of the sealers were analyzed at a 5% significance level.

## 3. Results

### 3.1. FTIR Analysis

The FTIR spectra of MTA-Bioseal and MTA-Fillapex showed a similar composition (Figure 1 and Table 1). Spectra of both sealers showed sharp bands for calcium hydroxide (Ca(OH)_2_); a broad band of the hydroxyl ion (OH) of absorbed water; bands of methyl (C–H) (Figure 1A) and amide I (C=O) from salicylate resin; and bands for carbonate groups (CO_3_^2−^), sulfate (SO_4_^2−^), and silicate groups (Si–O) from tri-calcium silicate (C_3_S), di-calcium silicate (C_2_S), and/or calcium silicate hydrate (CSH) (Figure 1B). In addition, the spectra of MTA-Bioseal showed phosphate bands (v_3_ and v_4_ PO_4_^3−^). The spectra of Adseal showed amide I (C=O), carbonate (CO_3_^2−^), silicate (Si–O), and phosphate (v_3_ and v_4_ PO^3−^) bands (Figure 1 and Table 1).

### 3.2. Solubility %

Both of the MTA-based sealers demonstrated an increased mean solubility % over time, with no significant difference (*p* > 0.05) between them. Adseal showed a significantly lower solubility % compared with the MTA-based sealers (*p* < 0.001) (Figure 2A).

### 3.3. pH Changes

As shown in Figure 2B, the MTA-Fillapex incubation solution underwent a rapid pH increase after the first day, reaching 10.25. The pH of the solution decreased to 9.73 after 28 days. The MTA-Bioseal solution did not show a similar high pH value after one day. However, it gradually increased over time, and by day 28 it had plateaued at a significantly higher pH than the MTA-Fillapex solution (*p* < 0.001). At each observation point, the solutions of the two MTA-based sealers had higher pH levels than that of Adseal (Figure 2B).

### 3.4. Calcium, Phosphate, and Silicon Ions Released

After the first day, Adseal had released significantly more Ca^2+^ ions than either of the MTA-based sealers (*p* < 0.001); however, MTA-Fillapex released significantly more Ca^2+^ than the others throughout the remaining immersion period (*p* < 0.001). At the end of the experiment, Adseal was found to release the least amount of Ca^2+^ (*p* < 0.001) (Figure 2C).

The three sealers exhibited variable leaching patterns for PO_4_^3^^−^ (Figure 2D). MTA-Bioseal released the largest amount of PO_4_^3^^−^ after the first day (*p* < 0.001). It then showed a sharp decline in the registered quantity, followed by a sharp increase to become the highest PO_4_^3^^−^ releasing sealer by the end of the observation period (*p* < 0.001). MTA-Fillapex and Adseal demonstrated opposite patterns. The amount of PO_4_^3^^−^ released by MTA-Fillapex consistently decreased with time, while the amount released by Adseal consistently increased (Figure 2D).

Figure 2E shows that MTA-Fillapex consistently released the largest amount of Si^4+^ throughout the observation period, followed by MTA-Bioseal, while Adseal consistently released the least amount (*p* < 0.001).

### 3.5. SEM/EDX Analysis

#### 3.5.1. Characterization of Sealers before Solubility Test

The MTA-Bioseal exhibited a homogeneous surface structure with similarly sized globular particles (Figure 3A). The surface also exhibited grayish areas between the particles and scattered bright dots of radio-opacifiers.

The surface of MTA-Fillapex showed a homogeneous layer of differently shaped particles that were mainly globular (belite) with a few scattered, irregularly shaped particles (alite) and elongated, irregular, bright particles of bismuth oxide in between (Figure 3B).

The Adseal surface showed a uniform structure of irregular small particles and bright radio-opacifier particles. The surfaces of the three materials contained microporosities. The largest number of microporosities was recorded for Adseal (large number of small microporosities, ranging from 10 to 54 µm^2^ in size), followed by MTA-Bioseal (few microporosities, with large irregular spaces; porosities ranged from 54 to 83 µm^2^ in size). The fewest microporosities were seen on the surface of MTA-Fillapex (few porosities, with small sizes ranging from 2.7 to 11.4 µm^2^) (Figure 3A–C).

The EDX analysis revealed that the three sealers were composed of carbon (C), oxygen (O), silicon (Si), phosphate (P), and calcium (Ca) at different concentrations (Figure 3D–F). Ca was significantly higher in MTA-Bioseal (*p* < 0.001), whereas Si was significantly higher in MTA-Fillapex (*p* = 0.003). Both of the MTA-based sealers contained aluminum (Al), whereas sulfur (S) was only detected in MTA-Bioseal. MTA-Fillapex and Adseal contained the same radio-opacifier (bismuth (Bi)), while MTA-Bioseal contained strontium (Sr) and titanium (Ti).

#### 3.5.2. Characterization of Sealers after Solubility Test

After being stored in deionized water for 28 days, the MTA-Bioseal surface showed a collapse of the micropores (Figure 3G), whereas the pores of MTA-Fillapex and Adseal became fewer in number and larger in size, ranging from 180 to 455 µm^2^ and from 8.4 to 18.9 µm^2^, respectively (Figure 3H,I).

EDX analysis revealed a decrease in the amount of Si on the surface of the two MTA-based sealers and an increase in Ca in all three sealers (Figure 3J–L).

### 3.6. Flow/Film Thickness

MTA-Fillapex registered the lowest mean flowability value (19.5 ± 0.35 mm), which was significantly lower than that of MTA-Bioseal (22.1 ± 0.42 mm) and Adseal (21.0 ± 0.61 mm) (*p* < 0.001). The two MTA-based sealers exhibited a similar thickness (50 µm), while Adseal showed a significantly greater thickness (130 ± 30 µm) (*p* < 0.001).

## 4. Discussion

The current study evaluated several physicochemical properties of MTA-Fillapex, MTA-Bioseal, and Adseal, as well as the effect of solubility on their surface morphology and composition. To the best of our knowledge, no previous studies have evaluated MTA- Bioseal. Our results showed that the three sealers differed in composition, degree of solubility, pH change in the surrounding medium, and type and concentration of released elements. The sealers also showed changes in their surface after being immersed in deionized water for 28 days. They also differed in their film thickness and flowability. Hence, the null hypothesis was rejected.

According to manufacturer information, MTA-Bioseal and MTA-Fillapex are composed of salicylic resin, 40% MTA (C_3_S, C_2_S, tricalcium aluminate, and calcium oxide), and radio-opacifiers [3,8]. The current study was the first to evaluate the properties and composition of MTA-Bioseal. The chemical composition of MTA-Fillapex has been extensively investigated [25,26,27]. The current EDX analysis detected C, O, Al, Si, P, and Ca in both of the MTA-based sealers (Figure 3D,E). This finding was in line with several previous studies [25,26,28,29]. To reduce cytotoxicity, the newer MTA-Bioseal included a lower Al percentage of ≈0.05 wt% instead of ≈0.2 wt% in MTA-Fillapex [30]; it also contained Sr and Ti as radio-opacifiers instead of Bi, which is in MTA-Fillapex [31]. Cell viability is significantly decreased in the presence of Bi, while the opposite was true for Sr [31]. The addition of Ti to the sealer has been previously reported and is owing to its effective antifungal properties [32]. Phosphorus is added to enhance the bioactivity and apatite formation of MTA-based sealers [33]. The FTIR spectra of MTA-Bioseal and Fillapex showed similar compositions with variable intensity bands. Their spectra showed bands for Ca(OH)_2_, methyl (C–H), and carbonyl (C=O) groups of salicylate resin and carbonate (CO_3_), sulfate (SO_4_), and SiO_4_ groups of calcium silicate hydrate (CSH). There were small bands for polymerized silicate (CxS), di-calcium silicate (C_2_S), and/or tri-calcium silicate (C_3_S). The phosphate (v_3_ and v_4_ PO_4_^3−^) bands were more prominent in the spectra of MTA-Bioseal, indicating its prominent bioactivity.

Sealer solubility is unfavorable because when a sealer disintegrates in the surrounding tissues it can lead to inflammatory and cytotoxic reactions [29]. According to international standards (ISO 6876 and ANSI/ADA Specification No. 57), the solubility of root canal sealers should not exceed 3% mass fraction when stored in water [6,34,35,36].

The results showed that the MTA-based sealers had similar high solubility that gradually increased over time and exceeded >4% mass fraction by the end of the observation period (Figure 2A). Several studies have reported a high solubility for MTA-Fillapex [7,36,37,38,39,40] that exceeds the acceptable limit [36]. However, other studies have detected lower weight loss values ranging from 0.25% after 28 days [41] to 4.65% after 6 months [6]. Such variability could be attributed to sealer shrinkage after immersion in water [42], excessive disintegrated elements leaching into the aqueous medium [43], or instability of the sealer matrix upon hydration with more soluble incorporated additives [36]. The current study found an increase in the leachable amount of Si^4+^ (Figure 2E) and a reduction in Si wt% on the surface of both MTA-based sealers after 28 days of water immersion (Figure 3J,K), which supports these suggestions [43]. In addition, the presence of hydrophilic particles on the surface of the MTA-based sealer allows more water molecules to encounter the sealer, thereby increasing its solubility [34].

Adseal is an epoxy resin-based sealer containing calcium phosphate, amines, and bismuth subcarbonate [44,45]. It was chosen as a control because of its resistance to solubility [29,46,47]; this resistance could explain why Adseal displayed a negative solubility % (Figure 2A). Adseal showed reduced leaching of Ca^2+^ and Si^4+^ compared to the MTA-based sealers, which might be attributable to elements being more thoroughly incorporated within the matrix during material polymerization. Adseal gained weight, which could be attributed to the susceptibility of the resin-based sealer to water sorption and a high expansion potential during and after polymerization [29,48]. Increased solubility was observed for Adseal by day 28. This can be attributed to the disintegration and breakdown of unreacted polymerized particles [49].

Changes in pH have been related to the degree of solubility and the amount of Ca^2+^ released [6,38,46]. The pH changes were attributed to the formation of calcium hydroxide during the hydration reaction followed by its dissociation into OH^−^ and Ca^2+^ [46]. This was confirmed by the significantly higher Ca^2+^ release for the two MTA-based sealers, with a maximum value at 21 days (Figure 2C). In the current study, both MTA-based sealers showed the highest mean value on day 7. The same pH value for the MTA-Fillapex solution was previously recorded in several studies [6,25,38,39,42], while a lower pH value (7.7–9.39) was recorded by others [27,50]. This suggests that pH changes are related to time [38]. Among the experimental periods, the MTA-Bioseal solution showed a higher pH value (Figure 2B) than that stated by the manufacturer (pH = 9) [8].

The high alkalinity of the MTA-based sealer solutions may be attributed to the pozzolanic reaction and the formation of Ca(OH)_2_ during the hydration reaction. Ca(OH)_2_ dissociates into OH^−^ and Ca^2+^, which promote antibacterial ability and osteogenic potential, respectively [6,51,52,53]. However, the prolonged alkalinity of the MTA-based sealer solutions might be considered as a source of cytotoxicity, leading to protein destruction and enzymatic cell membrane denaturation [54]. This adverse effect could be of clinical concern, as our results showed an increase in Ca wt% on the sealers’ surfaces (Figure 3J,K). Such accumulation might lead to cytotoxic events. A higher calcium content released by MTA-Fillapex compared with epoxy resin has been previously reported [40,50]. Siboni et al. reported that the maximum calcium content released by MTA-Fillapex was detected within the first 3 days, while epoxy resin (AHplus) did not exhibit calcium release at all [27]. The Adseal solution was weakly alkaline (Figure 2B), in contrast to the nearly neutral pH (≈7.5) [29] or acidic pH (≤6.5) [55] previously recorded. The lower pH changes induced by Adseal might be related to its lower solubility and reduced Ca^2+^ release.

The prolonged release of Ca^2+^, PO_4_^3^^−^, and Si^4+^ results in degradation of the sealer’s surface. SEM/EDX analysis revealed a decrease in the Si wt% (Figure 3D,E,J,K), with large micropores detected in both of the MTA-based sealers. MTA-Fillapex released a significant amount of Ca^2+^ and Si^4+^. This has been previously reported [52]. MTA-Bioseal demonstrated the greatest PO_4_^3^^−^ release (Figure 2C–E), which might be due to its higher P content compared with MTA-Fillapex, as detected by EDX. The marked increase in PO_4_^3−^ by the end of the observation period might have been due to its lack of attachment within the set sealer. This amount of PO_4_^3^^−^ release might enhance its bioactivity [33]. Conversely, Adseal showed greater PO_4_^3^^−^ release, which increased over time. This finding might be due to its greater P content, as detected by EDX.

The FTIR analysis identified PO_4_^3^^−^ in the spectra of MTA-Bioseal and Adseal at 1086 and 1031 cm^−1^, respectively [16,20,22]. The largest PO_4_^3^^−^ release was exhibited by MTA-Bioseal, followed by Adseal (Figure 2D). It appeared that PO_4_^3^^−^ was not well incorporated into the CSH structure of MTA-Bioseal; hence, it was easily released into the aqueous medium. Adseal is mainly composed of calcium phosphate [44]; thus, after polymerization, PO_4_^3^^−^ became incorporated within the sealer and was slowly released. The presence of PO_4_^3^^−^ within the sealer seems to enhance its bioactivity. This finding corroborated previous results [56].

The surfaces of the three materials contained microporosities, with the largest sizes in MTA-Fillapex. MTA-Fillapex exhibits a homogeneous surface with various sizes of porosities [38]. This may be related to the setting characteristics and the formation of a polymerized silicate phase [37]. Previous reports have shown that MTA-Fillapex is unable to set, even after 1 month [29]. Here, the FTIR spectra confirmed the presence of unhydrated calcium silicate (C_3_S and C_2_S) particles and little polymerized calcium silicate (CxS), with a low content of polymerized calcium silicate (CSH) [29]. Furthermore, there was low intensity of the SiO_4_ band at 900–800 cm^−1^ [19]. The presence of unhydrated silicate phase is responsible for the excessive Si^4+^ release and for the large micropores on the MTA-Fillapex surface. It is assumed that these micropores can hold water from the surrounding environment, allowing bacterial colonization [57]. Whether this would impact the long-term outcome of endodontic treatment warrants further clinical investigation.

Although improved flowability facilitates a sealer’s penetration into canal irregularities, excessive flow has been considered as a risk factor for extrusion and can potentially provoke inflammatory and cytotoxic reactions [41]. According to the ISO standard, the three tested sealers met the adequate flow specification (>17 mm) [10], with MTA-Bioseal registering the highest flowability, followed by Adseal and MTA-Fillapex. Previous studies have reported a wide range of MTA-Fillapex flowabilities (22–34 mm) [29,36,41,58]. Such high flowability could be due to a prolonged setting time [25,37] or a high resin/MTA ratio when used from freshly opened tubes [41].

Regarding film thickness, both MTA-Bioseal and Fillapex complied with the ISO standard (50 µm) [10], but Adseal had a high value (130 ± 30 µm). Previous studies have reported thick films for MTA-Fillapex (75 ± 12 µm) [36,42] and Adseal (0.083 mm) [44]. The flowability and film thickness of sealers may be influenced by their composition, small particle size, and setting characteristics [58]. The greater film thickness of Adseal can be attributed to its expansion after polymerization [29,48].

## 5. Conclusions

The three sealers differed in their composition, degree of solubility, induced pH changes in the surrounding medium, type and concentration of released elements, surface changes upon immersion in deionized water over 28 days, film thickness, and flowability. The two MTA-based sealers exhibited high solution alkalinity and released a considerable amount of Ca^2+^, which is conducive to osteogenic behavior. The greater solubility and Si^4+^ release exhibited by MTA-Fillapex might have led to the development of large micropores on its surface, which would compromise the apical sealing of the root canal system. This could be a clinical concern jeopardizing the long-term outcome of root canal treatment. Hence, clinicians should maximize efforts to limit contact of MTA-Fillapex with the surrounding periapical tissues. Further investigations are needed to evaluate the setting characteristics of MTA-based root canal sealers.

Despite the meticulous approach adopted in this study, the lack of moist conditions provided by dentinal tubule fluids, which aids in the setting reaction of MTA-based sealers, limits the extrapolation of our results to the clinical setting.

## Figures and Tables

**Figure 1 materials-14-05911-f001:**
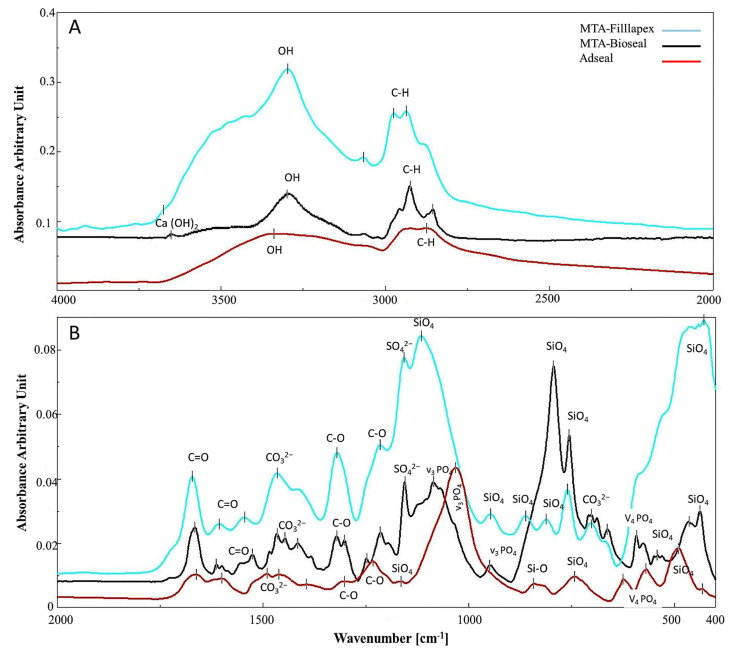
FTIR spectra of hydrated root canal sealers showing the composition of each sealer. At region 4000–2000 cm^−1^ (**A**), the spectra detected bands of calcium hydroxide (Ca(OH)_2_)_,_ hydroxyl ion of absorbent water (OH), methyl (C–H). At region 2000-400 cm^−1^ (**B**), the spectra detected amide I (C=O) of salicylate resin, carbonate (CO_3_^2−^), sulfate (SO_4_^2−^), phosphate (PO_4_), and silicate group (SiO_4_) of calcium silicate.

**Figure 2 materials-14-05911-f002:**
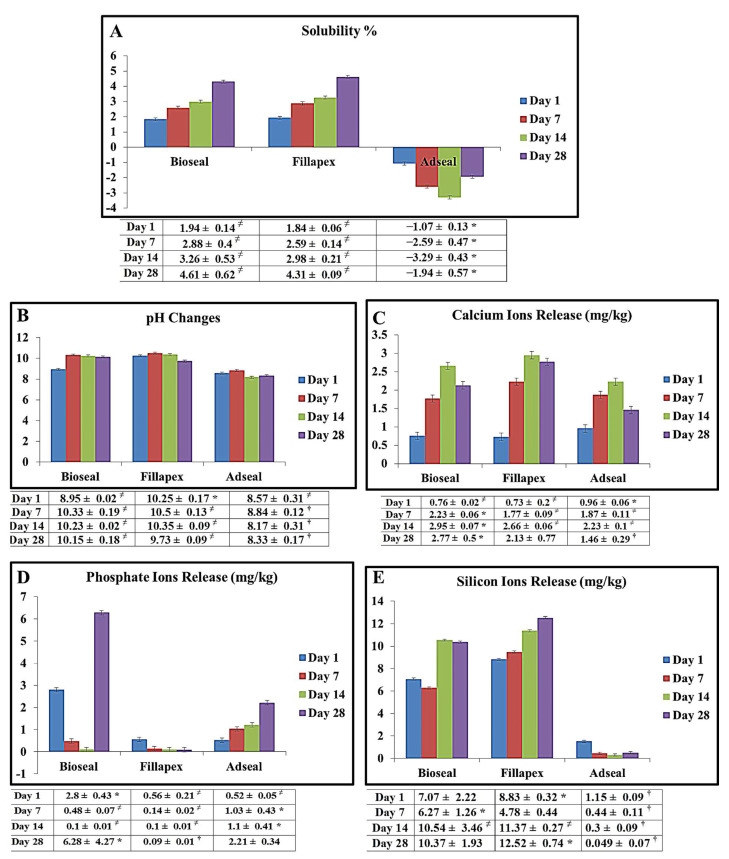
Histograms representing the mean values of solubility % (**A**), pH changes (**B**), Ca^2+^ ion release (**C**), PO_4_^3−^ ion release (**D**), and Si^4+^ ion release (**E**) of the root canal sealers over the immersion times of the experiment. * indicates the highest significant value (at *p* < 0.001). ^†^ indicates the lowest significant value (at *p* < 0.001). ^≠^ indicates no significant difference between sealers of the same symbol (*p* > 0.05).

**Figure 3 materials-14-05911-f003:**
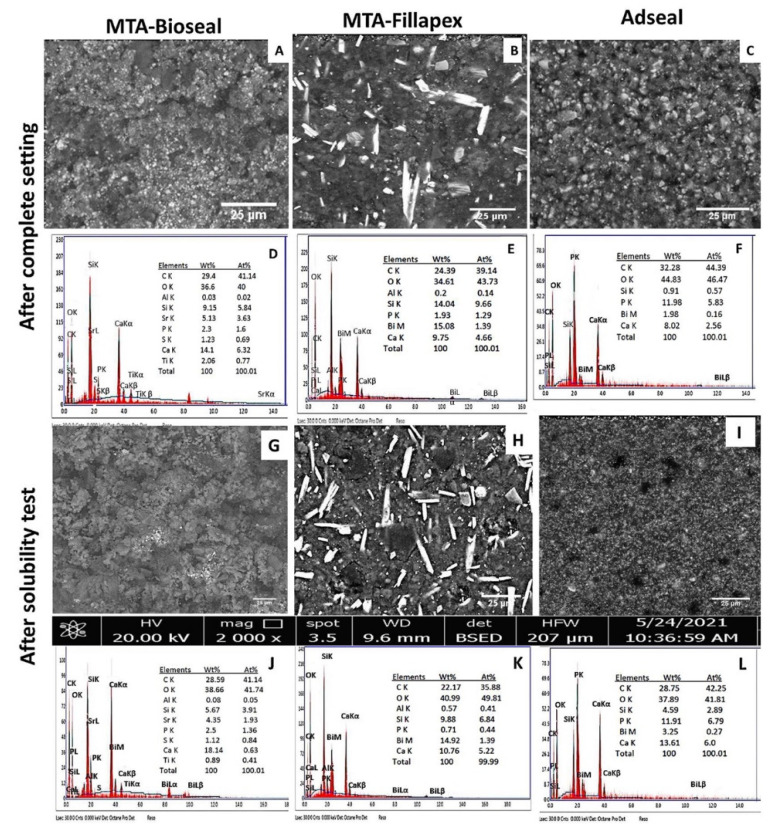
SEM examination of surface structure (**A**–**C**) and EDX analysis (**D**–**F**) of MTA-Bioseal, MTA-Fillapex, and Adseal before immersion in deionized water. SEM examination of surface structure (**G**–**I**) and EDX analysis (**J**–**L**) of MTA-Bioseal, MTA-Fillapex, and Adseal after immersion in deionized water for 28 days. Magnification ×2000.

**Table 1 materials-14-05911-t001:** Fourier transform infrared spectroscopy (FTIR) information of analyzed root canal sealers.

MTA-Bioseal (cm^−1^)	MTA-Fillapex (cm^−1^)	Adseal (cm^−1^)	Assignment (Vibration Mode) (Reference)
3642	3641		Ca(OH)_2_ [16,17,18,19]
3292	3298		OH [19]
2954, 2923, 2854, 1321, 1315	2972, 2935, 2873, 1318		CH [18]
1666	1671	1631	C=O of amide I [20]
1465, 1446	1464	1458	CO_3_^2−^ [16,17,19,20]
1321	1320	1303	CO [20,21]
1215	1248, 1215	1246	C–O of aromatic [21]
1155	1157	1165	SO_4_^2−^ [17,19]
	1112		SiO_4_ [18]
1086		1031	V_3_PO [16,20,22]
950	947		Si–O of calcium silicate hydrate (CSH) [19]
	860, 815		Si–O of lowly polymerized silicate (CxS) [18,23] (445 + 815 + 950 = C–S–H)
795, 75,710	760	710, 673	symmetric stretching of v_4_ SiO_4_ of CSH [19]
701	701, 690		CO_3_ of aragonite [24]
		618, 568	v_4_PO [22]
592			SiO_4_^2−^ bending of C_3_S [16,17,18]
465	464	500	SiO_4_^2−^ bending of C_2_S [18]
440	428	412	O–Si–O of CSH [17,18]

## Data Availability

Data available in a publicly accessible repository.

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
