# Peer review of "Physicochemical Properties of Two Generations of MTA-Based Root Canal Sealers"

_materials, 2021, doi:10.3390/ma14205911_

Round 1
Reviewer 1 Report
Authors made a compare with three commercial products. A series of physio-chemical properties were tested. The data were analyzed but a lot of questions need be improved.
- According to the basic components of the cements is a hydrolyzed and pozzolanic process in water, the chemical transformation should be tested and analyzed before and after curing.
- The reason why the changes of pH value and released elecments happened should be analyzed in details based on the chemical process.
- Some statistical analysis is not in right way. P=0.000 is better to be marked as different level of significant difference, such as p<0.05, p<0.01, p<0.001.
- In Figure 2, the different statistical difference should be marked as *, **, and *** between different group. It is too simple and not clear as the present state. And the tested sample number should be noted.
- In 3.3., the description of “Both MTA-based sealers maintained higher pH levels compared to Adseal throughout the observation period” is wrong.
- In 3.4., what is the reason of “a sharp decline in the registered quantity followed by peak increase to become the significantly highest releasing sealer over the three groups at the end of the observation period (P=0.000).”
- In 4. Discussion, the description of “Sr and Ti as radio-pacifiers instead of Bi in MTA-Fillapex [29]” is not right because there is no Ti has been presented in the reference. Please check it.
Author Response
Moderate English changes required
- English editing was done
Authors made a compare with three commercial products. A series of physio-chemical properties were tested. The data were analyzed but a lot of questions need be improved.
- According to the basic components of the cements is a hydrolyzed and pozzolanic process in water, the chemical transformation should be tested and analyzed before and after curing.
- We agree with the reviewer. However, this was the subject of another research project we are doing including setting reaction, chemical characterization and phase composition of the three sealers.
2-The reason why the changes of pH value and released elements happened should be analyzed in details based on the chemical process.
- It was added in line 712-716 “It could be attributed to the formation of calcium hydroxide during the hydration reaction, then its dissociation into OH- and Ca2+(Huang 1998). This was confirmed by the significantly higher release of Ca2+ by both MTA-sealers with maximum value at 21 days (figure 2C).”
3- Some statistical analysis is not in the right way. P=0.000 is better to be marked as different level of significant difference, such as p<0.05, p<0.01, p<0.001.
- This was corrected following the recommendation of the reviewer.
4- In Figure 2, the different statistical difference should be marked as *, **, and *** between different group. It is too simple and not clear as the present state. And the tested sample number should be noted.
- This was corrected following the recommendation of the reviewer.
5- In 3.3., the description of “Both MTA-based sealers maintained higher pH levels compared to Adseal throughout the observation period” is wrong.
- Kindly recheck figure 2B with its table: pH values for both MTA-based sealers has been always higher than Adseal at each observation period. We changed the sentence into: Both MTA-based sealers maintained higher pH levels compared to Adseal at each observation period (Lines 439-440).
6- In 3.4., what is the reason of “a sharp decline in the registered quantity followed by peak increase to become the significantly highest releasing sealer over the three groups at the end of the observation period (P=0.000).”
- The explanation was added in discussion “However, MTA-Bioseal demonstrated the greatest PO43- release (Figure 2 C-E) that might be due its higher P content than MTA-Fillapex as detected in EDX analysis. At the end of observation period, marked increase of PO43- release might be due to lack of its attachment within cured sealer. It might enhance its bioactivity (Al-Sanabani et al 2013).” Lines 785-789.
7- In 4. Discussion, the description of “Sr and Ti as radio-pacifiers instead of Bi in MTA-Fillapex [29]” is not right because there is no Ti has been presented in the reference. Please check it.
- Yes we agree with the reviewer. It was already described in discussion and we added the explanation related to Ti “Cell viability was significantly decreased in the presence of Bi, while the opposite was true for Sr [29]. The addition of Ti to the sealer was previously reported due to its effective antifungal properties (Raura et al 2020)” (Lines 671-673)

Reviewer 2 Report
The plagiarism has been checked and it showed a low acceptable level of similarity.
Abstract
- Kindly rewrite the abstract without headings.
Introduction:
- The Authors affirmed to evaluate the physico-chemical properties of two MTA-based sealer but important characteristics of these sealers such as radiopacity and setting time were not addressed. It could be interesting evaluate the new sealer in these terms, it will enrich the investigation quality. However, if Authors do not want to evaluate these two characteristics, kindly rearrange the aim of the study specifying which properties they decided to evaluate.
Materials and Methods:
This section has been well conducted and written.
- Kindly explain in a more detail the SEM Observation conditions used ( i.e. accelerating, working distance)
- Why were the study design and protocol reviewed and approved by an ethical committee if it is an in vitro study?
Results are clearly described
Discussion:
- kindly discuss the results comparing them with those of other published study obtained from similar testing conditions (obviously related to the 2 sealers previously investigated).
- Kindly add a paragraph on the limitation of the study and its clinical significance.
Author Response
The plagiarism has been checked and it showed a low acceptable level of similarity.
- iThenticate report recorded 1% plagiarism and idnetifed terms such as “Root canal sealer, Root canal system, tricalcium silicate, dicalcium silicate, scanning electron microscopy and energy dispersed X-ray, manufacture name and address of sealers used” as source of plagiarism. These terms was excluded. Other sentences identified as sourse of plagiarism have been rephrased.
Abstract
- Kindly rewrite the abstract without headings.
- This was done.
Introduction:
- The Authors affirmed to evaluate the physico-chemical properties of two MTA-based sealer but important characteristics of these sealers such as radiopacity and setting time were not addressed. It could be interesting evaluate the new sealer in these terms, it will enrich the investigation quality. However, if Authors do not want to evaluate these two characteristics, kindly rearrange the aim of the study specifying which properties they decided to evaluate.
- The radiopacity test was not available, while the setting time was addressed in another study including setting characterization and will be published later by the same authors. The aim of study was adjusted as the following “The current study aimed to evaluate the physico-chemical properties (including solubility%, pH changes, released calcium, phosphate and silicon ions, flow and film thickness) and the effect of solubility on surface morphology and composition of two generations of MTA-based root canal sealers”
Materials and Methods:
This section has been well conducted and written.
- Kindly explain in a more detail the SEM Observation conditions used ( i.e. accelerating, working distance)
- details were added to figure 3
- Why were the study design and protocol reviewed and approved by an ethical committee if it is an in vitro study?
- These are internal regulations in our institute in order to allow access to the research lab.
Results are clearly described
Discussion:
- kindly discuss the results comparing them with those of other published study obtained from similar testing conditions (obviously related to the 2 sealers previously investigated).
- All properties was discussed with the manuscript as shown in track change version
- Kindly add a paragraph on the limitation of the study and its clinical significance.
- We added that within the conclusion paragraph
- However, greater solubility and Si4+ release exhibited by MTA-Fillapex might develop large micropores on its surface that compromises apical sealing of the root canal system. This could be a clinical concern jeopardizing the long-term outcome of root canal treatment. Hence, clinicians should maximize efforts to limit contact of MTA-Fillapex with the surrounding periapical tissues. Further investigations are needed to evaluate the setting characterization of MTA-based root canal sealers.
- Despite the meticulous approach adopted in the study, the lack of moist conditions provided by the dentinal tubules fluids, which might aid in the setting reaction of MTA-based sealers, might limit the extrapolation of our results to the clinical setting.

Round 2
Reviewer 1 Report
Authors did some works and the results were rational. The paper is good for published now.